# Novel Variant in *PLAG1* in a Familial Case with Silver–Russell Syndrome Suspicion

**DOI:** 10.3390/genes11121461

**Published:** 2020-12-05

**Authors:** Yerai Vado, Arrate Pereda, Isabel Llano-Rivas, Nerea Gorria-Redondo, Ignacio Díez, Guiomar Perez de Nanclares

**Affiliations:** 1Rare Diseases Research Group, Molecular (Epi) Genetics Laboratory, Bioaraba Health Research Institute, Araba University Hospital-Txagorritxu, Vitoria-Gasteiz, 01009 Araba, Spain; yerai.vadoranedo@osakidetza.eus (Y.V.); arrate.peredaaguirre@osakidetza.eus (A.P.); 2NanoBioCel Research Group, Laboratory of Pharmacy and Pharmaceutical Technology, Faculty of Pharmacy, Universidad del País Vasco/Euskal Herriko Unibertsitatea (UPV/EHU), Vitoria-Gasteiz, 01006 Araba, Spain; 3Service of Genetics, BioCruces Health Research Institute, Hospital Universitario Cruces, Barakaldo, 48903 Bizkaia, Spain; isabel.llanorivas@osakidetza.eus; 4Service of Paediatric Neurology, Araba University Hospital, Vitoria-Gasteiz, 01009 Araba, Spain; nerea.gorriaredondo@osakidetza.eus; 5Department of Pediatric Endocrinology, Bioaraba Health Research Institute, Araba University Hospital-Txagorritxu, 01009 Vitoria-Gasteiz, Spain; IGNACIO.DIEZLOPEZ@osakidetza.eus

**Keywords:** Silver–Russell syndrome, *PLAG1*, *HMGA2-PLAG1-IGF2* pathway

## Abstract

Silver–Russell syndrome (SRS) is a rare growth-related genetic disorder that is mainly associated with prenatal and postnatal growth retardation. Molecular causes are not clear in all cases, the most common ones being loss of methylation on chromosome 11p15 (≈50%) and maternal uniparental disomy for chromosome 7 (upd(7)mat) (≈10%). However, pathogenic variants in genes such as *CDKN1C, HMGA2, IGF2,* or *PLAG1* have also been described. Previously, two families and one sporadic case have been reported with *PLAG1* alterations. Here, we present a case of a female with clinical suspicion of SRS (i.e., intrauterine and postnatal growth retardation, triangular face, psychomotor delay, speech delay, feeding difficulties). No alterations in methylation or copy number were detected at chromosomes 11p15 and 7 using methylation-specific multiplex ligation-dependent probe amplification (MS-MLPA). The custom panel study by next-generation sequencing (NGS) revealed a frameshift variant in the *PLAG1* gene (NM_002655.3:c.551delA; p.(Lys184Serfs *45)). Familial studies confirmed that the variant was inherited from the mother and it was also present in other family members. New evidence of pathogenic alterations in the *HMGA2-PLAG1-IGF2* pathway suggest the importance of studying and taking into account these genes as alternative molecular causes of Silver–Russell syndrome.

## 1. Introduction

Silver–Russell syndrome (SRS, OMIM#180860) is a rare imprinting genetic disorder that is mainly associated with intrauterine (IUGR) and postnatal growth retardation (PNGR) without catch-up. Initially, it was also defined by other features such as relative macrocephaly at birth, triangular face, body asymmetry, facial dysmorphic features, and severe feeding difficulties [1,2]. Subsequently, new case reports led to the inclusion of other signs, such as a low body mass index (BMI) or speech delay (for review, see Reference [3]). The incidence of the disease varies from 1/30,000 to 1/100,000 [3]. Most of SRS cases are sporadic, with few familial cases that suggest following an autosomal dominant inheritance pattern [4].

The wide variability of the clinical manifestations of SRS has led to the international recommendation about the use of the Netchine–Harbison clinical scoring system (NH-CSS) [5], both for determining when SRS genetic testing should be run and when a clinical diagnosis of SRS should be given (≥4 NH-CSS criteria, including both prominent forehead and relative macrocephaly) [3]. Although the molecular etiology cannot be determined in all cases, approximately 50% of the clinically diagnosed SRS patients present hypomethylation (Loss of Methylation, LOM) at 11p15.5 (*H19/IGF2*:IG-DMR), whereas 10% of them show maternal uniparental disomy for chromosome 7 (upd(7)mat) [3]. For the remaining 40% of patients, methylation-specific testing for 14q32 alterations, molecular karyotyping, and analysis of different tissues is advised [3,6,7,8], leading to the identification of underlying genetic alterations. However, in some cases, the underlying molecular mechanism is still unknown.

In addition, maternally transmitted *CDKN1C* activating variants in some families [9,10,11] and paternal *IGF2* inactivating alterations in some other cases [8,12,13,14,15,16,17,18] have been described. Moreover, sequence variants of two non-imprinted genes (*HMGA2* and *PLAG1*) have been associated with SRS. *HMGA2* variants have been described in three families and three sporadic cases [12,19,20] and *PLAG1* alterations in two families and in a sporadic case [12,21] (Appendix A).

*PLAG1* is a transcription factor that contains seven canonical C_2_H_2_ zinc finger domains for the DNA binding (expanding from residue 34 to 236) and a C-terminus with a serine-rich region that has transcriptional activation activity (residues 385–500). *PLAG1* binds the P3 promoter of *IGF2,* thereby increasing its expression*. IGF2* is known to be an essential growth factor for the normal embryonic growth and its gene is maternally imprinted [22]. Its overexpression can also lead to uncontrolled cell proliferation [23].

Here, we describe a novel variant in *PLAG1* in a family with short stature and phenotypic manifestations compatible with SRS.

## 2. Materials and Methods

Informed consent for genetic analysis was obtained from patients or parents. The study protocol was approved by the ethical committee of the Basque Country (Code: PI2017018).

### 2.1. Case Report

We present the case of a 9-year-old-girl who is the first of two children of consanguineous parents (index’s grandmothers are first cousins) referred because of significant proportionate short stature (height 115 cm (below first percentile (*p*); −2.31 SDS), weight 18.4 kg (*p*5; −1.61 SDS), and head circumference (HC) 46.8 cm (*p* < 1)).

After spontaneous conception, intrauterine growth restriction, gestational hypertension, gestational diabetes (controlled by diet), and a small placenta were detected during pregnancy, so delivery was induced at 38 gestational weeks and she was born small for her gestational age (SGA) (birth weight 1600 g (*p* < 1; −4.24 SDS), birth length 41 cm (*p* < 3; −4.03 SDS), and HC 27 cm (*p* < 1; −5.61 SDS)).

In the postnatal period, growth restriction (PNGR) was also detected with weight and length under p3 during her evolution (Figure 1). She was reviewed for short stature for the first time aged 7 years by our Pediatric Endocrinology team (weight: 14.9 kg (*p*3–10; −1.38 SDS), length 107.2 cm (*p*2; −1.99 SDS)). Within the clinical history, retarded psychomotor development was suspected because she did not crawl and started to walk at 18 months. She spoke her first words near her second year of age, and at school she had problems with language development. An intelligence quotient test at 5.5 years revealed a total value of 61 with 58 (*p* < 3; <−2 SDS) for verbal abilities, as well as 72 (*p*3) for manual skills. She had no sleeping or behavioral problems, but she showed feeding difficulties.

She was referred to the clinical geneticist who observed the presence of a triangular face, subtle superficial hemangioma in the frontal midline, thin hair, long forehead, bulbous small nose, prominent chin, high arched palate with crowded teeth, and a fine voice. Neither asymmetry, organomegaly, hyperlaxity, nor other skin alterations were detected.

Her mother was 34 years old, with a height of 140 cm (*p* < 3; −2.64 SDS), and she mentioned having presented problems at school with learning difficulties. The mother has a narrow face, hypotelorism, thin hair, long forehead, crowded teeth, and clinodactyly of the fifth finger on both hands. When interviewed, she noted being short in stature all her life, yet she never associated with being low weight. Some of her siblings and her mother presented similar signs (Figure 2).

The father was also 34 years old and was 158 cm in height (*p* < 3; −2.15 SDS), with subtle rhizomelic shortening within the upper limbs and no dysmorphic features. No alterations were identified at the *SHOX* gene.

### 2.2. Molecular Genetic Studies

Genomic DNA of the patient and family members was extracted from peripheral blood or buccal swabs using the QIAamp DNA Mini Kit (QIAGEN, Hilden, Germany), following the manufacturer’s instructions.

Dosage and methylation analyses for chromosomes 11 (including *H19/IGF2*:IG-DMR and *KCNQ1OT1*:TSS-DMR loci), 6, 7, and 14 were carried out using methylation-specific multiplex ligation-dependent probe amplification (MS-MLPA) with ME030-C3 and ME032-A1 kits (MRC-Holland, Amsterdam, The Netherlands) respectively, following the manufacturer’s recommendations. Data was analyzed using Coffalyser.net version 9.4 software (MRC-Holland).

Subsequently, molecular karyotyping was performed using a 400 K microarray-based comparative genomic hybridization (aCGH) kit (Agilent Technologies, Santa Clara, CA, USA). Slides were scanned on an AgilentSureScan C Microarray scanner and analyzed with ADM-2 software.

For the screening of alterations in other differential diagnostic genes of SRS, our group designed and validated a custom panel of 26 genes/regions related to growth impairment. The study of this panel was conducted using Nextera Flex for the enrichment method (Illumina Inc., San Diego, CA, USA) and sequenced on a MiniSeq Mid Output kit with 2 × 150 cycles, 2.4 Gb (Illumina Inc.). The BaseSpace BWA Enrichment (Illumina Inc.) was used to process data (alignment with BWA 0.7.7 on GRCh37/hg19 and VCF and bam/bai files were obtained with GATK Variant Caller v1.6-23 and SAMtools v0.1.19, respectively). Downstream bioinformatic analysis of VCF files (annotation, filtering, and variant prioritization) and bam files (for CNV detections) was performed with the help of commercial software VarSeq V2.2.1 (Golden Helix, Bozeman, MT, USA). The parameters used for the filtering and prioritization were as follows: quality assessment (Q > 30; mean coverage > 50×), population frequency (minor allele frequency < 1% in ExAC, gnomAD, 1000 Genomes, or ESP6500 population databases), variant effect (missense, nonsense, frameshift, and splicing effect), and in silico pathogenicity prediction (Combined Annotation Dependent Depletion, CADD_1.4 score > 14). The resulting variants were classified in accordance with the American College of Medical Genetics and Genomics’ (ACMG) criteria [24].

After filtering, variant confirmation, and co-segregation studies in other family members were carried out via Sanger sequencing using Bigdye Terminator version 3.1 kit and analyzed in an ABI 3500 (Applied Biosystems, Foster City, CA, USA), according to the manufacturer’s protocols.

## 3. Results and Discussion

### 3.1. MS-MLPA Study

Methylation analysis of chromosome 11 showed that the patient did not present variations in the methylation status nor in the copy number of the studied region (loci *H19*, *KCNQ1*, and *CDKN1C*). Indirectly, it was also demonstrated that there was not a maternal disomy.

MS-MLPA analysis of chromosomes 6, 7, and 14 revealed no methylation alteration for chromosomes 7 and 14, excluding upd(7)mat, as well as Temple syndrome, which were also observed in some patients with clinical features suspicious of SRS [25,26]. Curiously, a partial LOM in *PLAGL1*:alt-TSS-DMR was identified in the proband (Figure 3A) without dosage alteration, but not in her parents. LOM at *PLAGL1*:alt-TSS-DMR is the cause of 6q24-related transient neonatal diabetes mellitus [27] associated with fetal growth restriction [28]. *PLAGL1* plays an important role on the network of co-regulated imprinted genes involved in controlling fetal and post-natal development, hence variations in its expression may affect growth homeostasis. A study in a cohort of healthy subjects demonstrated that as *PLAGL1* expression is proportionally associated with its methylation status, small variations in methylation in the DMR and its subsequent higher expression of *PLAGL1* correlated with smaller fetal (from the second semester of pregnancy), birth, and infant weight, and BMI, but not with length (only in TDNM when LOM is total) [29]. In this particular case, the mother presented normal weight throughout her life, which correlated with her normal methylation status. No glucose alterations were detected in the proband, neither during the neonatal period nor later in her life. Besides, the mother, with the same phenotype, did not carry the PLAGL1 LOM, so it can be assumed that this alteration was not a direct cause of the clinical manifestations in the index, even if its influence in more severe manifestations, mainly during the prenatal and perinatal periods, cannot be excluded (lack of maternal data to compare).

### 3.2. aCGH Array

The 400 K aCGH array revealed a duplication on the short arm of chromosome 9, at 9p24.1(arr(hg19) (8,157,331*–*8,629,525) × 3), which involves *PTPRD* gene exons 10*–*43. This duplication was inherited from her mother. The PTPRD gene is involved in pre- and post-synaptic differentiation of neurons. There is no evidence correlating this alteration with any known syndrome or disease. Initially, we hypothesized that this alteration could co-segregate with learning difficulties in the family, but the grandmother (II-4) and the aunt (III-5) were not carriers, disproving this hypothesis (Figure 2). It may be a previously undescribed polymorphism, even if according to ACMG guidelines [24] it was classified as a variant of unknown significance.

### 3.3. Next-Generation Sequencing

After filtering, only one potentially causative variant remained: a heterozygous pathogenic variant at exon 5 of the *PLAG1* gene (NM_002655: c.551delA; p.(Lys184Serfs *45)). In particular, ACMG criteria [24] evaluated this variant as a null variant in a gene where loss of function (LOF) is a known mechanism of disease (LOUEF:0.19) (PVS1) absent in population databases (PM2), and in silico predictions support a deleterious effect (PP3). The presence of the variant was confirmed using Sanger sequencing (Figure 3B) and it was confirmed to be maternally inherited. This variant was also harbored by the grandmother and two of the aunts and uncles (Figure 2).

As previously reported in the literature, *PLAG1* pathogenic variants have only been described in three cases: two of them familial and the other one sporadic. In the three cases, the variants produced a truncated protein (two frameshift and a nonsense) and the disease followed a dominant transmission pattern [12,21], as was the case of our patient. All variants are located at the last coding exon of the gene, so a transcribed mutant mRNA will presumably not be degraded by a nonsense-mediated decay (NMD) pathway [30]. The aberrant proteins lead to a lack of zinc-fingers that could affect DNA binding. Specifically, in p.Lys184Serfs *45, as from residue 184, the protein frame is totally different, and the last two zinc fingers (F6 and F7) will disappear, as in the new protein, the C2H2-type motif [C-x(2,4)-C-x(3)-[LIVMFYWC]-x(8)-H-x(3,5)-H] [31] is totally lost. In fact, it has been demonstrated in vitro that when F6 and F7 are removed, the protein F1–F5 does not show any clear DNA binding, revealing the absolute requirement of the last two zinc fingers for the DNA binding [32]. Analysis of previously reported variants reveals that the p.(Ser147Valfs *82) [12] and p.(Arg197 *) [21] would probably have a similar effect because in both cases, F6 and F7 are removed. In p.(Gln455Serfs *16) [12], even if all the zinc fingers seem to be conserved, transcription activation could be affected as residues 385*–*500 are presumably important for this task. Furthermore, this region is conserved across species (Uniprot, Q6DJT9 entry).

It has been shown that there is a correlation between *PLAG1* and *IGF2* levels [12]. IGF2 is a growth factor and its expression is regulated by several genes, one of them being *PLAG1*, so the important role of the HMGA2-PLAG1-IGF2 pathway in controlling growth has been demonstrated. On the other side, as described by Abi-Habib et al., silencing *PLAG1* resulted in lower levels of total IGF2-P3 [12], which should be highly active in fetal tissues [24]. The putative low expression of *IGF2* during fetal development could explain the intrauterine growth restriction observed in these patients with *PLAG1* haploinsufficiency [12].

Keeping in mind the differences already pointed out by Abi-Habib et al. [12], we reviewed the SRS NH-CSS six key features in patients with alteration in the *HMGA2-PLAG1-IGF2* pathway (see Appendix A). The characteristic SGA and PNGR were present in all patients, although PLAG1 patients seem to be the less affected [12,21]. In fact, as Abi-Habib et al. and Inoue et al. observed, the birth length SDS of *PLAG1* patients is around *−*2.5 instead of *−*4, which is the mean of SDS for patients with alterations in *HMGA2, IGF2*, or *CDKN1C* [12,21]. However, the index cases of the present report presented an SDS for birth length similar to that observed for *HMGA2, IGF2*, or *CDKN1C*. Further data are needed to obtain conclusive information. The relative macrocephaly at birth (defined as a difference in SDS of at least 1.5 between head circumference and length or weight at birth) seems to vary, even inside the same family, since some patients have been reported as showing microcephaly, as in the index of the present family [20]. The protruding forehead and feeding difficulties (and/or low body mass index) appear to be very prevalent in this group with genetic alterations [12,15,20,21]. Finally, the body asymmetry, as expected, was not described in this group of patients with genetic germline variants (except for 3 patients with IGF2 alterations [12,16]) as this is usually a characteristic of a mosaic distribution of 11p15 LOM [21].

Apart from the typical SRS characteristics (e.g., IUGR, PNGR, feeding difficulties), the index and her affected relatives also showed dental problems, comprehension/intellectual difficulties, and fine hair. Dental problems in SRS patients have been previously described in the literature, and were also reported in two patients with *HMGA2* alterations (not previously mentioned in patients with *PLAG1* alterations) [12,33]. However, regarding comprehension/intellectual difficulties, even though several reports on SRS children mentioned a reduced intelligence quotient (IQ) [34], a recent study carried out in adults with the typical 11p15 epimutation came to the conclusion that the IQ score was in the normal range and the patients had no problem with verbal comprehension [35]. Learning difficulties were present in the current family both in carriers and non-carriers of the PLAG1-variant, but unfortunately, IQ testing was just done for the index. Therefore, we cannot univocally correlate the variant with learning difficulties. Nevertheless, intellectual developmental delay has been reported in two non-related patients with *IGF2* alterations (IQ/DQ = 77 and 79) [15], and attention-deficit hyperactivity disorder and difficulties were mentioned in a recently reported patient with a *PLAG1* nonsense variant [21]. Further studies on this aspect are needed.

## 4. Conclusions

Firstly, as previously described in a few patients, we presented the case of a girl with clinical features of SRS with a frameshift pathogenic variant in the *PLAG1* gene. It is the fourth truncating variant identified in this gene associated with SRS. All of them seem to affect its ability for DNA binding. *PLAG1* is a regulator of growth, so it should be taken into account in SRS molecular diagnosis and, maybe, in other conditions affecting growth.

Secondly, *HMGA2, PLAG1,* and *IGF2* are genes involved in the same pathway. All of them have been associated with growth, so studying the *HMGA2-PLAG1-IGF2* pathway in cases of SRS, once the most common alterations are excluded, could be a useful approach.

Finally, the *HMGA2-PLAG1-IGF2* pathway should be considered as a new genetic cause of Silver–Russell syndrome because it can lead to growth restriction.

## Figures and Tables

**Figure 1 genes-11-01461-f001:**
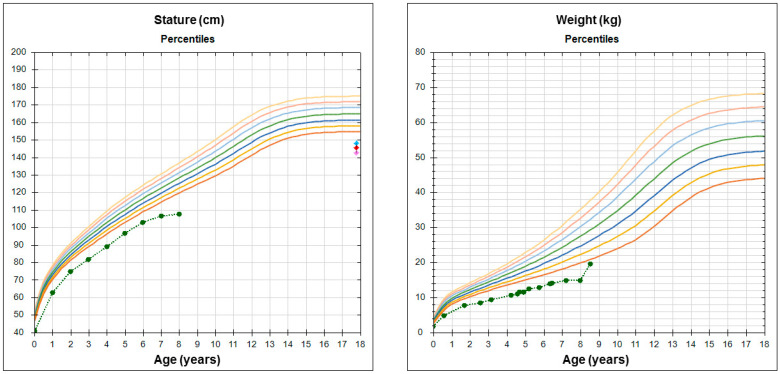
Height and weight charts. Green dots and dashed line represent the patient’s data.

**Figure 2 genes-11-01461-f002:**
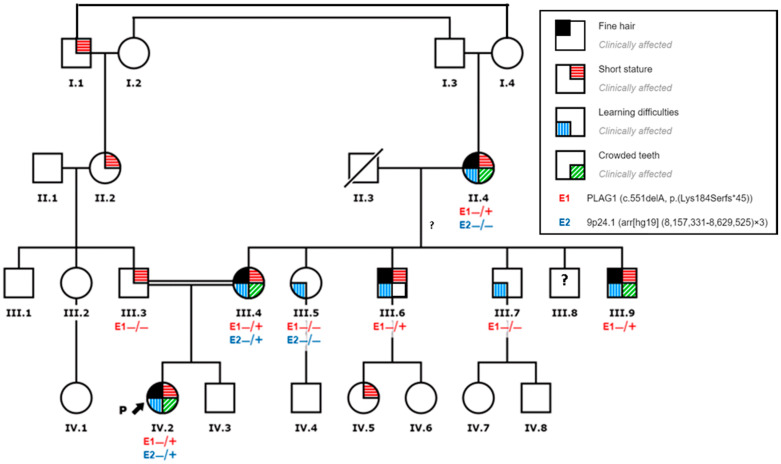
Family pedigree: the arrow points to the index case. The family members carrying the variant are represented as (+/−) versus non-carriers (−/−). The presence of each feature is represented by a colored square.

**Figure 3 genes-11-01461-f003:**
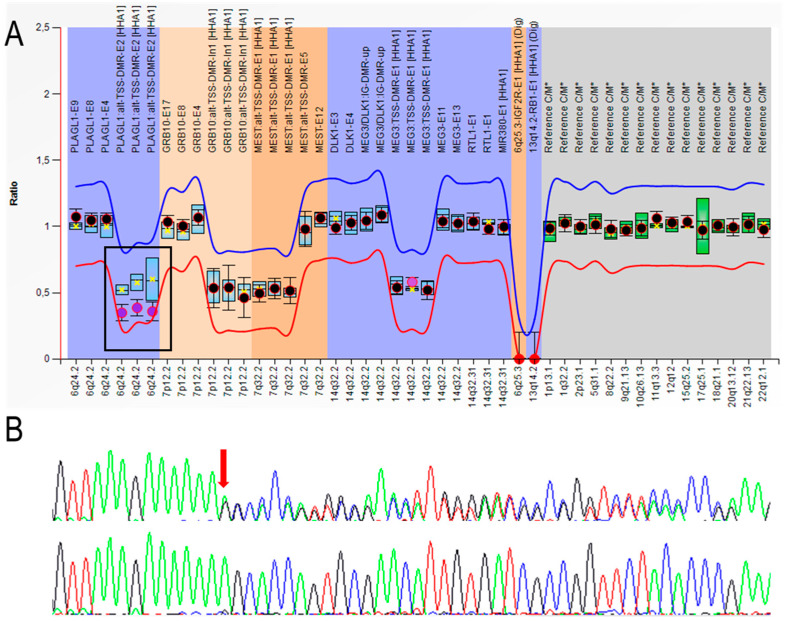
(**A**) Partial hypomethylation in *PLAGL1:*alt-TSS-DMR (framed) revealed in ME032-A1 kit analysis, represented with Coffalyser. Probe results indicate significantly decreased signals compared to the reference samples as a decrease of more than two standard deviations. The red and blue lines indicate the arbitrary borders for loss and gain, respectively (by default, the borders are placed ±0.3 from the mean probe value of a probe over the reference samples). Black dots display the probe ratios of the patient’s sample, and the error bars the 95% confidence ranges. The blue and green box plots in the background show the 95% confidence range of the used reference samples, the yellow star indicates the mean. When a probe ratio crosses the borders, it is indicative for a gain or loss (black dot surrounded in red) of methylation. Full red dots correspond to internal controls of *HhaI* digestion. (**B**) Electropherogram of the confirmatory direct sequencing of identified pathogenic variant in the *PLAG1* gene. Upper panel belongs to the proband, showing NM_002655: c.551delA; p.(Lys184Serfs *45) heterozygous variant (arrow) and the lower one is a wild-type sequence.

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
