# Peer review of "Novel Variant in PLAG1 in a Familial Case with Silver–Russell Syndrome Suspicion"

_genes, 2020, doi:10.3390/genes11121461_

Round 1

Reviewer 1 Report

The manuscript is sound. This is an interesting report of a family with features of SRS that will make a contribution to the biomedical literature. The manuscript is clearly written although some of the English is a little quirky. I have made some editorial marks on the manuscript (see attached). In addition there are a few sentences that are not quite clear enough.

Author Response

We thank the reviewer for his/her kindly comments and his/her help with English editing.

Regarding the questions included in the text:

  • Line 74, p is for percentile; it has been explained
  • Line 86, we have detailed the percentiles for the different intelligence quotient tests
  • Line 132, ACMG acronym has been explained
  • Figure 3: we have included more detailed data within the legend for figure explanation
  • Lines 189 and 197: reference’s errors have been solved
  • Lines 216, 218 and 220: confusing phrases has been edited to be clearer

Reviewer 2 Report

The manuscript by Vado and colleagues reports on a case of Silver-Russell syndrome. The authors excluded the most common causes of SRS, i.e. hypomethylation of the IGF2-H19 DMR at 11p15 or maternal UPD7. They also found no mutation in SRS candidate genes IGF2, CDKN1C, and HMGA2. Interestingly, they identified a novel mutation in the PLAG1 gene and proposed this is the cause of the disease. This mutation is only the 4th PLAG1 mutation reported in SRS patients, and adds to our knowledge of the disease.

1- The proband displayed features typical of SRS patients, e. g. reduced birth weight, reduced postnatal height, and feeding difficulties, which resulted in an elevated NH-CSS. Other typical features such as body asymmetry were missing and atypical features such as supernumerary teeth, fine hair... were observed. It is not quite clear to me if the patient is a bona fide SRS patient or if, as stated in the conclusion (… we present the case of a girl with SRS suspicion…), the SRS diagnosis is arguable. Its looks like the SRS phenotype is intermixed with other defects not typical of SRS patients. The title of the manuscript should be modified accordingly.

2- The identification of a truncating mutation in PLAG1 is highly suggestive in the present context (SRS suspicion with no LOM at 11p15 and no matUPD7). The authors should comment on the structure of the mutant gene. The truncation occurred near the end of the zinc finger region. How many zinc finger domains were conserved in the mutated protein? Were the conserved/deleted zinc finger domains known to be involved in DNA binding? Could the truncated protein behave as a dominant negative for the wt protein? How does this mutation compare to PLAG1 mutations previously identified?

3- The authors observed other (epi)genetic defects in the proband; a duplication of exon 10-43 of PTPRD, which was also present in the proband’s mother, and a partial demethylation of the PLAGL1 promoter, which was not present in the mother.

The author should comment on the expected impact of PTPRD partial duplication on the gene product. What protein is expected to be produced from the rearranged gene(s)?

The demethylation of PLAGL1 is suggested not to be involved in the observed phenotypes. One argument is that the mother did not display the same PLAGL1 demethylation and was affected (...the mother, with the same phenotype, did not carry the PLAGL1 LOM, so it can be assumed that this alteration was not a direct cause of the clinical manifestations…). The supplementary table lists the phenotypes displayed by the index and her mother; no phenotype, but ‘final height’, was determined for the mother. The authors should clarify if the mother was indeed affected, and to which extent.

The second argument was that the observed partial loss of PLAGL1 promoter methylation did not result in alteration of PLAGL1 expression levels. First, this should be documented in the supplemental data. Which tissue was used to measure PLAGL1 expression? If PLAGL1 expression was determined in PBL, the experiment is not conclusive as the gene is bi-allelically expressed at low levels in blood cells. The effect of the promoter demethylation on mono- vs. bi-allelic expression is reliably observed in cells with a much higher PLAGL1 levels, e.g. fibroblasts. In the same paragraph, the authors suggest that the level of PLAGL1 promoter methylation predicts PLAGL1 expression levels. They cite reference 30 to support this assertion. An erratum was published last summer which invalidated the correlation between PLAGL1 promoter methylation status and PLAGL1 expression in females. The correlation still held in males but the index is a girl.

Figure 3A is very obscure to me. What are the blue and red lines? What variable is represented by the box-plots? Is the IGF2R-E1 region at 6q25.3 also demethylated?…

4- Line 64. ‘IGF2 is known ... and its gene is maternally imprinted [22]’. IGF2 is paternally imprinted (the methylation mark is acquired on the paternal allele in the male germline) and paternally expressed.

5- The conclusion section is weak.

6- The manuscript is generally well written. Yet, some sentences require editing. Just a few examples.

Line 20. ‘...not clear in all cases, being the most common ones loss of methylation...’ → ...not clear in all cases, the most common ones being loss of methylation…

Line 58. ‘… three family...’ → families

Line 72. ‘9 years-old-girl’ → 9-year-old girl

Please have the manuscript checked by a native English reader.

Author Response

Thank you for your review that has helped us to improve our manuscript.
